# Birth Weight < 3rd Percentile Prediction Using Additional Biochemical Markers—The Uric Acid Level and Angiogenesis Markers (sFlt-1, PlGF)—An Exploratory Study

**DOI:** 10.3390/ijerph192215059

**Published:** 2022-11-16

**Authors:** Magdalena Bednarek-Jędrzejek, Sylwia Dzidek, Piotr Tousty, Ewa Kwiatkowska, Aneta Cymbaluk-Płoska, Tomasz Góra, Bartosz Czuba, Andrzej Torbé, Sebastian Kwiatkowski

**Affiliations:** 1Department of Obstetrics and Gynecology, Pomeranian Medical University, 70-204 Szczecin, Poland; 2Department of Nephrology, Transplantology and Internal Medicine, Pomeranian Medical University, 70-204 Szczecin, Poland; 3Department of Gynecological Surgery and Gynecological Oncology of Adults and Adolescents, Pomeranian Medical University, 70-204 Szczecin, Poland; 4Clinical Department of Gynecology and Obstetrics, John Paul 2nd Municipal Hospital, 35-241 Rzeszow, Poland; 5Department of Obstetrics and Gynecology, Medical University of Silesia, 40-055 Ruda Slaska, Poland

**Keywords:** birth weight < 3rd percentile, uric acid, angiogenesis biomarkers, low birth weight, fetal growth restriction

## Abstract

(1) Aim: Ultrasound is the gold standard for assessing fetal growth disorders. The relationship between high sFlt-1/PlGF scores and LBW (low birth weight) was described. In this study, we attempted to assess whether uric acid could be used as a secondary marker in estimating the pregnancy risk associated with LBW. (2) Material and methods: 665 pregnant women with a suspected or confirmed form of placental insufficiency were enrolled. In each of the patients, sFlt-1 and PlGF and uric acid levels were determined. Patients were divided into two groups according to birth weight below and above the third percentile for the given gestational age with the criteria of the neonatal definition of FGR (fetal growth restriction). (3) Results: A significant negative correlation between neonatal birth weight and the uric acid level across the entire study group was observed. We found a significant negative correlation between neonatal birth weight and the uric acid level with birth weights < 3rd percentile. (4) Conclusions: There is a significant link between the uric acid concentration and LBW in the group with placental insufficiency. Uric acid can improve the prediction of LBW. An algorithm for LBW prognosis that makes use of biophysical (ultrasound) and biochemical (uric acid level, angiogenesis markers) parameters yields better results than using these parameters separately from each other.

## 1. Introduction

Low birth weight (LBW) is a serious health problem present not only in developing countries. In those regions, maternal malnutrition and underweight, as well as inadequate care during pregnancy, are the principle causes of this gestational complication [1]. In highly developed countries, however, preterm labor is one of the main causes, followed by fetal growth disorders, which affect up to 10% of pregnancies [2,3]. The reasons for growth disorders are multi-factorial, not only maternal but also fetal (e.g., chromosome aberrations), although up to 30% of the cases are due to placental insufficiency [2].

Intrauterine growth restriction entails a multitude of complications in both early and adult life. It is the second major cause of perinatal mortality [4]. It may cause neonatal complications such as hypothermia, hypoglycemia [5], and increased susceptibility to infection immediately after birth [6], as well as serious conditions in adult life, i.e., type 2 diabetes, hypertension, and metabolic syndrome [7,8]. Many attempts at treating the condition have been developed, but so far, the best therapy has been by rapid identification of pregnancies carrying an increased risk of fetal growth restriction (FGR) and their monitoring in order to capture the appropriate time for planned delivery.

Ultrasound is the gold standard for assessing fetal growth disorders [9]. Gordijn et al. have attempted to determine the necessary parameters to diagnose the FGR syndrome. Fetal biometry and appropriate Doppler ultrasound flows are used in diagnosing midgestational FGR according to the Delphi method [10]. A team of experts has also created criteria for the neonatal definition of FGR [11]. They claim that growth disorders in a newborn can be recognized if the birth weight is <3rd percentile (PC) or if three of the following characteristics are present: body weight < 10th percentile, head circumference < 10th percentile, body length < 3rd percentile, perinatal FGR diagnosis, and information obtained from the mother during pregnancy.

Presently, an array of compounds is being investigated for a potential correlation with low birth weight further on in the pregnancy. An abnormally functioning placenta produces angiogenic and anti-angiogenic factors in a disturbed manner. Among these factors, the most important are soluble fms-like tyrosine kinase-1 (sFlt-1), also referred to as soluble VEGF receptor-1 (VEGFR-1), and placental growth factor (PlGF). These are the only placental insufficiency markers identified and used in clinical practice so far. German and UK recommendations treat these parameters as standard in diagnosing preeclampsia [12]. Specific sFlt-1/PlGF ratio values have a strong predictive value in diagnosing a developing preeclampsia. Our previous publications showed correlations between sFlt-1/PlGF ratio values and perinatal outcomes in pregnancies that required a determination of this marker [13,14]. The relationship between high sFlt-1/PlGF scores and low birth weight was described. Particular attention should be given to the correlations between low placental growth factor levels and low birth weight [15].

During all our studies of placental insufficiency, we carefully evaluate the clinical and biochemical parameters of pregnancies, requiring sFlt-1/PlGF determinations, i.e., those suspected of having any of the forms of placental insufficiency. In addition to the markers that we discussed in our previous publications, we have noticed important correlations between birth weight and uric acid levels. When analyzing the available literature, we have come across reports of a relationship between different uric acid levels and low birth weight [16,17].

In the present study, we attempted to assess whether uric acid could be used as a secondary marker in estimating the pregnancy risk associated with low birth weight in pregnancies requiring an sFlt-1/PlGF determination, i.e., those with a suspected or confirmed form of placental insufficiency.

## 2. Materials and Methods

The study included 665 pregnant women hospitalized in the Clinical Department of Obstetrics and Gynecology of the Pomeranian Medical University of Szczecin between May 2017 and June 2020 with a suspected or confirmed form of placental insufficiency. The qualification was based on criteria for identifying individual disease entities in accordance with the applicable guidelines:Gestational hypertension, i.e., hypertension diagnosed beyond 20 weeks of gestation (wkGA), with values exceeding 140/90.Preeclampsia, i.e., hypertension with proteinuria or renal or hepatic impairment, and hematological disorders, as well as the clinical signs of uteroplacental dysfunction in the form of intrauterine death or FGR.HELLP syndrome, i.e., the occurrence of such signs as hemolysis, increased aminotransferase levels, and thrombocytopenia.Eclampsia, i.e., the occurrence of tonic-clonic seizures, is secondary to preeclampsia or not.FGR according to specific ultrasound criteria [10] after ruling out genetic causes (some patients had diagnostic amniocentesis performed, while others were evaluated after delivery).Placental abruption was diagnosed on the basis of clinical signs, bleeding, and uterine tetanus, and a final diagnosis was made postpartum.

Patients diagnosed with placental insufficiency were monitored according to generally accepted standards using biochemical, biophysical, and general condition parameters. Additionally, the angiogenesis markers sFlt-1 and PlGF, as well as uric acid levels, were determined in all patients.

Standardized reagent kits using the “ECLIA” electrochemiluminescent immunoassay and a colorimetric enzymatic assay were used for biochemical determinations.

Blood for the determinations was sampled upon each patient’s informed consent, and then centrifuged for 30 min. and stored at −80 °C awaiting analysis.

Upon delivery, birth weight and duration of gestation at birth were assessed, and based on these data, the Fenton growth charts were used to determine the corresponding birth weight percentiles [18]. The newborns were divided, respectively, into those with birth weights below and above the third percentile for the given gestational age in accordance with the criteria of the neonatal definition of FGR [11].

The research was conducted with the consent of the local ethics committee of the Pomeranian Medical University (no. n.KB-0012/122/12) and supplemented by a consent to the expansion of the research team (no. n.KB-0012/46/18).

The data obtained in the study were statistically analyzed using Statistica 13.1 by StatSoft. For the individual variables, the basic descriptive statistics—the mean, the standard deviation, and the median—were calculated. The minimum and maximum values were determined. Percentages were calculated for the descriptive, non-numerical variables. The significance level of *p* < 0.05 for all statistical differences was adopted. *p*-values are not adjusted for multiple tests and should be interpreted exploratorily only. A comparative analysis was performed between the different patient groups using the following statistical methods:The Shapiro–Wilk test was used to study data normality. Non-normal distributions of the evaluated parameters were observed.Due to the non-normal distributions, non-parametric tests were applied. The Mann–Whitney U test was used to identify the correlations between the parameters analyzed.The correlations between the compounds studied and the individual clinical, biochemical, and ultrasound parameters and the perinatal outcomes were compared using a Pearson correlation test and a significance test for regression coefficients.

## 3. Results

Our patients were evaluated according to the birth weight percentile criterion, i.e., by distinguishing between those with a neonatal birth weight < 3rd PC (n = 97) and >3rd PC (n = 568). In these groups, a multitude of significant differences were found, primarily between ultrasound parameters, the most significant ones being between the uterine artery and umbilical artery flow Doppler ultrasound measurements—these parameters are part of the definition of growth disorders proposed in 2016 by a team of experts according to the Delphi method [10]. Significant differences were also found between the angiogenesis markers and individual perinatal outcomes. The median values and the significance levels of these parameters are presented in the table below—Table 1.

Our analysis of the correlations between individual parameters and birth weight across the entire study group showed numerous significant relationships, especially with ultrasound and angiogenesis parameters. What drew our attention was the significant negative correlation between neonatal birth weight and the uric acid level (R = −0.193; *p* < 0.05). The individual correlations across the entire study group are shown in Table 2.

Given the above, we made a closer analysis of the group of patients with fetal growth disorders, i.e., those with birth weights < 3rd PC. The correlations between birth weight and the individual parameters, especially uric acid levels, are shown in Table 3.

Across the entire study group, significant positive correlations were shown between uric acid levels and abnormal Doppler ultrasound results for the umbilical artery flows (R = 0.126; *p* < 0.05), and significant correlations with the disturbed angiogenesis markers. As for the group of patients with neonatal birth weights < 3rd PC, uric acid levels significantly correlated with birth weight (R = −0.317; *p* < 0.05). The individual correlations with uric acid levels, both across the entire study group and within the group of patients with neonatal birth weights < 3rd PC, are shown in Table 4 and Table 5.

Linear regression analysis helped evaluate the effects of both ultrasound and biochemical parameters on birth weight. Table 6 shows the significance of the predictive capacity of the individual parameters in respect of low birth weight, both alone and in combination, which significantly increases the possibility of predicting this perinatal outcome.

Figure 1 shows linear regression line between birth weight and uric acid levels in the <3rd PC group.

## 4. Discussion

The concept of FGR is difficult to define. Unlike SGA, FGR fetuses are not merely “small”, as they may even reach the correct weights as per growth charts. However, these are fetuses that have not reached their biological growth potential due to growth-restricting factors developing during pregnancy. Especially in the case of late-onset FGR, i.e., developing beyond 32 wkGA, it is extremely important to distinguish between FGR and SGA. Fetuses below the 10th percentile are very often small but healthy. This is why our study focused on newborns with a birth weight below the third percentile, which, according to both the midgestational ultrasound definition and the neonatal definition, are referred to as suffering from fetal growth restriction.

For a number of years, different methods have been sought to help diagnose insufficiently large fetuses at risk of intrauterine death. In the 1970s, fetal growth disorders were associated with the mother’s history, i.e., her obstetric diseases, chronic diseases, and dependence on alcohol or cigarettes [19,20,21]. Attempts were also undertaken to measure fetuses using the biparietal diameter of the fetal head in ultrasound [22,23] or symphysial fundal height (SFH) [24], or evaluating total intrauterine volume [25]. Correlations between fetal weight and placental lesions were noticed. An accurate morphological evaluation of the placentae was also commenced following low birth weight deliveries [26]. Additionally, attempts were undertaken to evaluate the variability of the biochemical parameters in pregnancy complicated by growth disorders. In such cases, lower levels of estradiol, pregnanediol, human chorionic somatomammotropin, and heat-stable alkaline phosphatase (HSAP), among others, were found [23,27]. However, knowledge of the agents useful in diagnosing FGR has changed over the years. In the 1980s, it was considered, for example, that the PAPP-A protein level was not helpful and could not be used in the diagnosis of fetuses with growth disorders [28,29]. In turn, nowadays, i.e., since around 2003, it has been considered that a low level of PAPP-A during the first-trimester screening correlates strongly with low birth weight [30,31,32]. Ultrasound is currently the primary tool in the prediction of low birth weight. The criteria defined by the team of experts allow the diagnosis of FGR fetuses [10]. However, they are still not perfect as far as predicting low birth weight. That is why compounds are still being sought that will help diagnose this condition. Pedroso et al. have shown that the flows in uterine arteries alone are not sufficiently accurate in predicting fetuses with growth disorders [33]. Abnormal flows in uterine arteries and the umbilical artery are often evidence of placental dysfunction and damage. They have made their way into the practice of diagnosing the different forms of placental insufficiency, and their usefulness is also confirmed by our study results. However, they are not always sufficiently effective. Therefore, researchers are still in the search for such molecules that will facilitate a rapid diagnosis of low birth weight.

A few of them have appeared promising in this respect. In our reports, we have focused on demonstrating the value of the angiogenesis markers. FGR as diagnosed by neonatologists denotes neonates with birth weights < 3rd PC. Our research confirmed the relationship between a birth weight < 3rd percentile and the angiogenesis markers. The sFlt-1/PlGF ratio is also used in the prediction of serious placental abnormalities, which is why attempts have been undertaken to use it and determine its value in the pathology discussed here as well. Visan, Sovio, and Gaccioli have proven in their publications that the ability to diagnose fetuses demonstrating intrauterine growth restriction and their inferior perinatal outcomes is greatly increased by adding the sFlt-1/PlGF ratio values to the maternal risk factors and the ultrasound parameters that allow for diagnosing fetal growth disorders [34,35,36]. In our previous publication, we proved lower birth weights in the group of patients with the highest sFlt-1/PlGF values [13]. In the current study, the highest sFlt-1/PlGF values were found in the group of patients with birth weights < 3rd PC. We showed that there are significant correlations between the impaired angiogenesis markers and birth weight. Ultimately, a linear regression analysis helped us prove that among all the biochemical parameters evaluated, the sFlt-1/PlGF ratio had the largest effect on low birth weight. Anderson has proven strong correlations between a low PlGF level and low birth weight, especially in fetuses with late-onset growth disorders [15]. In our study, we too were able to demonstrate strong correlations between birth weight and PlGF levels and also that, in combination with other biochemical and ultrasound parameters, PlGF levels have a significant effect on the ability to predict low birth weight.

Our analysis of the various parameters revealed their specific correlations with the uric acid level. It has been shown in physiological pregnancies that uric acid levels initially decrease until approximately 24 wkGA, and subsequently increase until delivery to reach values higher than those prior to pregnancy. This is due to an increased glomerular filtration rate and decreased reabsorption in the proximal tubule during the first half of pregnancy and decreased uric acid clearance resulting from secondary absorption of uric acid during the second half of pregnancy [37,38]. Higher uric acid levels have been associated with the presence of oxidative stress, which may have additional relevance to the growth of the fetus. Hyperuricemia is strongly associated with endothelial cell dysfunction. Uric acid stimulates the production of vasoconstrictive and inflammatory factors, reduces the production of nitric oxide, and increases the production of thromboxane in vascular smooth muscle cells. In Ryu, Nair, Kumar, and Escudero, uric acid has been shown to have a high prognostic value in the diagnosis of low birth weight in pregnancies complicated by preeclampsia and arterial hypertension [39,40,41,42]. Elevated uric acid concentrations may also have an inflammatory effect on small blood vessels in the placenta, which may contribute to the development of fetal growth disorders. Akahori et al. have shown that increased uric acid levels are strongly associated with low birth weight in pregnant women with normal arterial pressure values [16]. Zhou et al. have shown, in turn, that low birth weight may occur in cases with either extremely high or very low concentrations of uric acid [17]. As early as 1998, Merviel et al. proved that lower birth weight was the only identified effect of isolated hyperuricemia in children born to mothers diagnosed with isolated hyperuricemia lasting for over two weeks [43]. Our study showed strong negative correlations between the uric acid level and birth weight. These correlations were even more significant in the group of patients with birth weights < 3rd PC. In addition, uric acid demonstrated strong correlations with the impaired angiogenesis markers and the PI values in the umbilical artery as shown in Doppler ultrasound studies, i.e., parameters whose effects on low birth weight prognosis have been proven. Using linear regression analysis, we showed that uric acid levels in our patients were an important prognostic factor for low birth weight. This is in line with the latest research published by Sun et al. [44]. In their study on a group of 7995 pregnant patients, strong negative correlations between uric acid levels and birth weight were shown, thus leading to the conclusion that the serum uric acid concentration can be a reliable marker for predicting adverse pregnancy outcomes, especially low birth weight. Multiple regression models can be used to show that when evaluated jointly, the parameters discussed above greatly increase our ability to predict low birth weight. A combination of parameters such as UA-PI, UtA-PI, sFlt-1/PlGF, and the uric acid level has the largest prognostic value in respect of low birth weight in mothers with a suspected or confirmed form of placental insufficiency. Each of the above-mentioned markers on its own is an important prognostic factor for low birth weight, but using a combination of different ultrasound parameters and biochemical factors, primarily sFlt, PlGF, and the uric acid level, provides a better ability to predict low birth weight than ultrasound parameters alone.

## 5. Conclusions

There is a significant link between the uric acid concentration and low birth weight in the group of patients suffering from placental insufficiency.Uric acid is one of those markers that can improve the prediction of low birth weight.The application of an algorithm for low birth weight prognosis that makes use of biophysical (ultrasound) and biochemical (uric acid level, angiogenesis markers) parameters yields better results than using these parameters separately from each other.

## Figures and Tables

**Figure 1 ijerph-19-15059-f001:**
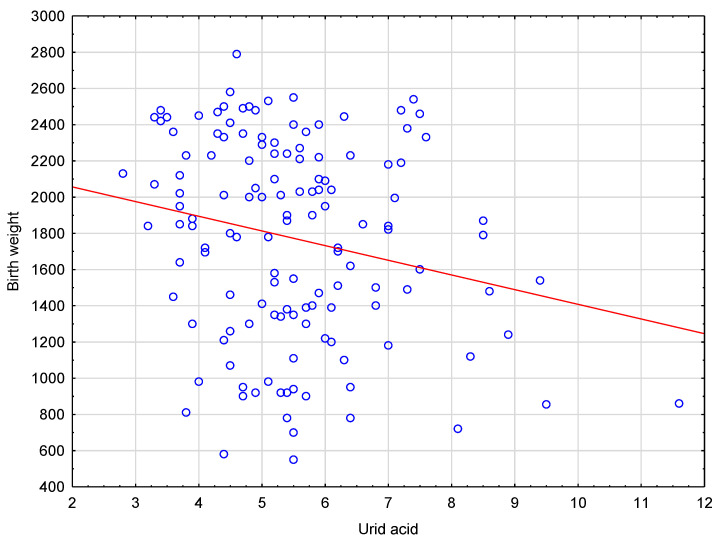
Scatterplot and a linear regression line between birth weight and uric acid levels in the <3rd PC group. The red line shows the regression line showing the relationship between birthweight < 3rd PC and uric acid level; blue hollow circles—patients who gave birth babies with birthweight < 3rd PC.

**Table 1 ijerph-19-15059-t001:** Characteristics of the study group.

	Birth Weight Percentile		
	<3 pc Median (Min–Max)	≥3 pc Median (Min–Max)	95% CI
**Parity**	n = 97	n = 568	1.529–1.723
	1 (1–7)	2 (1–9)	
**Gravidity**	n = 97	n = 569	1.27–1.422
	1 (1–6)	1 (1–7)	
**Weight, kg**	n = 85	n = 473	79.284–81.883
	73 (50.4–140)	78 (47–142)	
**Height, cm**	n = 87	n = 466	164.808–165.699
	164 (150–179)	165 (151–182)	
**UAPI**	n = 80	n = 366	0.707–0.82
	1.11 (0.56–4.59)	0.91 (0.51–7.48)	
**UtAPI**	n = 48	n = 164	0.699–0.839
	1.02 (0.36–2.94)	0.87 (0.22–2.51)	
**Mean arterial pressure**	n = 87	n = 494	103.96–106.459
	105 (72–150)	105.67 (65–159)	
**Uric acid, umol/L**	n = 85	n = 490	4.795–5.015
	5 (2.8–9.4)	5.1 (2.3–11.6)	
**AST, U/L**	n = 82	n = 508	20.561–26.044
	19 (10–128)	18 (8–807)	
**ALT, U/L**	n = 82	n = 508	19.075–24.369
	15 (5–159)	15 (3–574)	
**LDH, U/L**	n = 31	n = 348	204.045–214.18
	202 (140–296)	201 (118–1020)	
**PLT, ×10^9^/L**	n = 97	n = 568	215.021–224.426
	210 (91–421)	217 (57–445)	
**RBC, ×10^12^/L**	n = 97	n = 568	3.849–3.944
	4.12 (3.37–5.24)	4.16 (2.62–8.6)	
**Hb, mmol/L**	n = 97	n = 568	7.275–7.41
	7.6 (6–9.2)	7.6 (0.37–9.7)	
**Ht, %**	n = 97	n = 568	35.129–35.581
	0.35 (0.29–0.42)	0.36 (0.22–0.5)	
**WBC, ×10^9^/L**	n = 97	n = 568	10.579–11.027
	10.92 (5.79–17.82)	10.65 (4.32–31.55)	
**Fibrynogen, g/L**	n = 79	n = 438	4.162–4.351
	4.5 (2–7.1)	4.4 (1.6–16)	
**APTT, s**	n = 95	n = 548	26.821–27.193
	27.8 (21.4–32.6)	26.95 (20.4–40.9)	
**PT, s**	n = 93	n = 550	10.157–10.27
	10.3 (9.2–12.2)	10.5 (9.1–12.5)	
**D-dimers, ng/mL**	n = 27	n = 261	1597.849–1848.991
	1190 (381–3445)	1317 (246–10,000)	
**sFlt-1, pg/L**	n = 97	n = 568	5354.455–6057.637
	5513 (942–31,097)	4176.5 (215.1–31,877)	
**PlGF, pg/L**	n = 97	n = 568	178.992–219.485
	86.4 (14.59–994)	133 (7.23–2616)	
**sFlt-1/PlGF ratio**	n = 97	n = 568	75.07–97.598
	70.93 (1.96–816.93)	34.83 (0.5–1479)	
**Delivery week**	n = 97	n = 566	36.079–36.55
	35 (26–40)	38 (25–41)	
**Cord blood, pH**	n = 71	n = 477	6.999–7.075
	7.31 (7.13–7.49)	7.31 (6.8–7.53)	

**Table 2 ijerph-19-15059-t002:** Analysis of the correlations between birth weight and the evaluated parameters across the entire study group.

Parameter	Correlation Coefficient	*p*-Value
**Weight**	R = 0.393	<0.001
**Height**	R = 0.177	<0.001
**UA-PI**	R = −0.476	<0.001
**UtA-PI**	R = −0.553	<0.001
**Uric acid**	**R = −0.193**	<0.001
**AST**	R = −0.100	0.032
**LDH**	R = −0.208	<0.001
**RBC**	R = 0.131	0.002
**Fibrinogen**	R = 0.124	0.013
**PT**	R = 0.191	<0.001
**D-dimers**	R = 0.121	0.040
**Sflt-1**	R = −0.224	<0.001
**PlGF**	R = 0.358	<0.001
**Sflt-1/PlGF**	R = −0.375	<0.001
**Week of labor**	R = 0.749	<0.001

**Table 3 ijerph-19-15059-t003:** Analysis of the correlations between birth weight and the parameters evaluated within the group with neonates below the 3rd PC.

Parameter	Correlation Coefficient	*p*-Value
**UA-PI**	R = −0.414	0.019
**Uric acid**	**R = −0.317**	0.049
**Week of labor**	R = 0.718	<0.001

**Table 4 ijerph-19-15059-t004:** Analysis of the correlations between uric acid levels and the evaluated parameters across the entire study group.

Parametr	Correlation Coefficient	*p*-Value
**UAPI**	R = 0.126	0.040
**MAP**	R = 0.262	<0.0001
**AST**	R = 0.153	0.0001
**ALT**	R = 0.095	0.048
**LDH**	R = 0.248	<0.0001
**PLT**	R = −0.12	0.008
**PT**	R = −0.286	<0.0001
**Sflt-1**	R = 0.382	<0.0001
**PlGF**	R = −0.413	<0.0001
**sFlt-1/PlGF**	R = 0.487	<0.0001
**Week of labor**	R = −0.216	<0.0001
**Fetal weight**	**R = −0.193**	<0.0001
**pH**	R = −0.121	<0.018

**Table 5 ijerph-19-15059-t005:** Analysis of the correlations between uric acid levels and the evaluated parameters within the group with neonates below the 3rd PC.

Parameter	Correlation Coefficient	*p*-Value
**MAP**	R = 0.441	0.012
**Sflt-1**	R = 0.499	0.001
**PlGF**	R = −0.32	0.028
**Sflt-1/PlGF**	R = 0.479	0.002
**Fetal weight**	**R = −0.317**	0.049

**Table 6 ijerph-19-15059-t006:** Multiple regression analysis of the effects of ultrasound and biochemical parameters on birth weight.

Parameter	Coefficient of Determination R^2^	*p*-Value
**Mean UtA-PI**	R^2^= 0.31	*p* < 0.0001
**UA-PI** **Mean UtA-PI**	R^2^= 0.37	*p* < 0.0001
**UA-PI** **Mean UtA-PI** **PlGF**	R^2^= 0.43	*p* < 0.0001
**UA-PI** **Mean UtA-PI** **sFlt-1/PlGF**	R^2^= 0.45	*p* < 0.0001
**UA-PI** **Mean UtA-PI** **sFlt-1/PlGF** **Uric acid**	**R^2^ = 0.53**	*p* < 0.0001

## Data Availability

Not applicable.

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
