# Peer review of "Birth Weight < 3rd Percentile Prediction Using Additional Biochemical Markers—The Uric Acid Level and Angiogenesis Markers (sFlt-1, PlGF)—An Exploratory Study"

_ijerph, 2022, doi:10.3390/ijerph192215059_

Round 1

Reviewer 1 Report

  Thank you for asking me to provide a review of this article, which has a subject of high interest nowadays, as Fetal Growth Restriction continues to be a real problem during pregnancies, especially when it complicates with preeclampsya, gestational hypertension or other pathologies. 

   The main purpose of the analysis was to assess whether uric acid could be used as a secondary marker in estimating the pregnancy risk associated with low birth weight (LBW). The study was conducted on a number of 665 pregnant women, for a period of time between May 2017 and June 2020, which from my point of view is quite sufficient for this kind of study. SFlt-1, PIGF and uric acid levels were determined in each of the patients and the obtained results were strong enough to draw a conclusion. 

  Regarding the structure and accuracy of the phrases, the manuscript has indeed well structured information, with supported evidence and well structured phrases.

  The manuscript is original and well defined and so, the results provide an advance in current knowledge. The results are being interpreted appropriately and are significant, as well as are the conclusions, which are, of course, supported by the results. So the article is written in an appropriate way. 

  The study is correctly designed and the analysis is being performed at high standards, so the data are robust enough to draw the conclusion. 

  Surely the paper will attract a wide readership. 

  The English language is appropriate and well understandable and only has very few writting mistakes, which can easily be corrected, so that the article could be of highest quality.

  I only have a few things to add in the lines below, strictly regarding the writting techniques, but it is clear that the article is completely adequate and deserves to be published: 

Line 48: the line which begins with „Ultrasound...” should be well aranged in the page, to be in perfect line with the other paragraphs

Line 49: the necessary parameters, not „the parameters necessary”

Line 51: have also, not „has also”

Line 71: „,” between „pregnancies” and „requiring”

Author Response

Thank You very much for Your positive review. We changed those typos and those combinations of words.  I attach the corrected manuscript.

Reviewer 2 Report

With great interest I read your manuscript. My comments are as follows:

Please explain abbreviations like FGR, LBW, PC even when well known to most of the audience. Either in abstract or introduction.

 Abstract, line 31:

“ (3)Results:  The significant negative correlation between neonatal birth weight and the uric acid level across the  entire study group. The significant negative correlation between neonatal birth weight and the uric “ I suggest to add corresponding verbs making it easier to read like …  correlation between birth weight and the uric acid level across the  entire study group was observed …..

A lot of p-values have been calculated which are prone to the well-known multiple testing issue. Either tackle this issue by adjusting for it or adding in methods something like “…. P-values are not adjusted for multiple tests and should be interpreted exploratorily only ….” Otherwise you would present too many false positive results without letting the readership knowing it. Furthermore, please add “-an exploratory study” to the title.

Table 1: Please explain what the content of the brackets means. Q1,Q3? I strongly suggest to add the difference between the 2 groups its 95%CI  as an additional column and to make the comparison more easy. A 95% CI is much more intuitive than p-values are.

Chart 1: “Correlations between birth weight and uric acid levels in the < 3rd PC group” This regression line seems not proper to me. Firstly, obviously the distribution of the corresponding residuals must be quite skew and rather not normally distributed. The points below the line are far away and therefore have strong leverage and have properly have utterly big impact on the regression line. This makes the regression line quite questionable. Secondly, what does this plot tell us? An increase from 2.8 to 8 in urid acid is – very questionable – associated with a mean increase in birth weight by about 110g. But this finding rather pertains to birth weights around 2200 g but certainly not to the extreme birth weights below 1500g. This “predictability” pertains only to the mean though but not to the individual predictive ability which can be seen that the points are quite distant to the estimated regression line..

The authors mention “multiple regression” but do not explain how this regression model looks like, i.e. which independent variables are included?

In general, the many, many correlation values and R-squared values shown in the manuscript rather confuse. Furthermore, a correlation of like -0.193 in table 2 for urid acid is a strong sign that it is not able to predict properly, even -0.31 in table 3 – which is hardly significant even when not considering multiple testing-points to very small predictability of individual observations.

That said, I am not convinced at all that “There is a strong link between the uric acid concentration and low birth weight” as stated by the authors. The observed significant p-values which result from the partially big sample size are not a sign for itself for good predictability. P-values and predictability are two completely different properties.

In conclusion, I strongly suggest to drop at least some  of the results,  shorten the manuscript by it , concentrate more on urid acid and its predictability without concentrating on its p-value and in general let an experienced statistician go through statistics again.

Author Response

Reviewer 1:

Thank You very much for Your review. I really appreciate it.

Point 1: Please explain abbreviations like FGR, LBW, PC even when well known to most of the audience. Either in abstract or introduction.

Response 1: We explained all abbreviations in abstract and in introduction.

Point 2:  

Abstract, line 31:

“ (3)Results:  The significant negative correlation between neonatal birth weight and the uric acid level across the  entire study group. The significant negative correlation between neonatal birth weight and the uric “ I suggest to add corresponding verbs making it easier to read like …  correlation between birth weight and the uric acid level across the  entire study group was observed …..

Response 2: We changed the sentences

Point 3: A lot of p-values have been calculated which are prone to the well-known multiple testing issue. Either tackle this issue by adjusting for it or adding in methods something like “…. P-values are not adjusted for multiple tests and should be interpreted exploratorily only ….” Otherwise you would present too many false positive results without letting the readership knowing it. Furthermore, please add “-an exploratory study” to the title.

Response 3: We changed the title and added this sentence.

Point 4: Table 1: Please explain what the content of the brackets means. Q1,Q3? I strongly suggest to add the difference between the 2 groups its 95%CI  as an additional column and to make the comparison more easy. A 95% CI is much more intuitive than p-values are.

Response 4: We explained the content of the brackets.

The significance level α is the maximum probability that we will make a type 1 error, i.e. the probability of rejecting the null hypothesis even though it was true. Put simply, it is the probability that we will find that our confidence interval does not contain the parameter we are looking for, even though it did.

The confidence level is the percentage that tells us how accurate the confidence interval is, i.e. how often we are right. The higher the confidence level (closer to 100%), the more often we are right about the parameter estimate.

The relationship between these concepts is simple.

significance level: α

confidence level: 1 – α

 We consider the significance level α to be sufficient for this table, adding Cl 95% would reduce the readability of the table.

Point 5: Chart 1: “Correlations between birth weight and uric acid levels in the < 3rd PC group” This regression line seems not proper to me. Firstly, obviously the distribution of the corresponding residuals must be quite skew and rather not normally distributed. The points below the line are far away and therefore have strong leverage and have properly have utterly big impact on the regression line. This makes the regression line quite questionable. Secondly, what does this plot tell us? An increase from 2.8 to 8 in urid acid is – very questionable – associated with a mean increase in birth weight by about 110g. But this finding rather pertains to birth weights around 2200 g but certainly not to the extreme birth weights below 1500g. This “predictability” pertains only to the mean though but not to the individual predictive ability which can be seen that the points are quite distant to the estimated regression line..

Response 5: We changed the chart.

Point 6: The authors mention “multiple regression” but do not explain how this regression model looks like, i.e. which independent variables are included?

Response 6: Those independent variables are included:  UA-PI, Mean UtA-PI and sFlt-1/PlGF ratio.

Point 7: 

In general, the many, many correlation values and R-squared values shown in the manuscript rather confuse. Furthermore, a correlation of like -0.193 in table 2 for urid acid is a strong sign that it is not able to predict properly, even -0.31 in table 3 – which is hardly significant even when not considering multiple testing-points to very small predictability of individual observations.

That said, I am not convinced at all that “There is a strong link between the uric acid concentration and low birth weight” as stated by the authors. The observed significant p-values which result from the partially big sample size are not a sign for itself for good predictability. P-values and predictability are two completely different properties.

In conclusion, I strongly suggest to drop at least some  of the results,  shorten the manuscript by it , concentrate more on urid acid and its predictability without concentrating on its p-value and in general let an experienced statistician go through statistics again.

Response 7: We changed the conclusions. 

I attach the corrected manuscript.

Reviewer 2:

Thank You very much for Your positive review. We changed those typos and those combinations of words.  I attach the corrected manuscript.

Round 2

Reviewer 2 Report

"The significance level α is the maximum probability that we will make a type 1 error, i.e. the probability of rejecting the null hypothesis even though it was true. Put simply,  … We consider the significance level α to be sufficient for this table, adding Cl 95% would reduce the readability of the table." Thank you for your effort explaing me what Type 1 and a CI are. Still, a correctl verbal definition of a CI runs like "A 95% CI covers the unknown, true parameter with a probability of 95%" and not "The confidence level is the percentage that tells us how accurate the confidence interval is, i.e. how often we are right. ".

Being statistician for many years I am quite aware of the compatibility between statistical tests and CIs. Nevertheless, as e.g. some known journals are requesting the use of confidence intervals only, a confidence interval enables a much better interpretation than a p-value does since it allows to quantify the effect. Table 1 for sure allows adding corresponding CIs for the differences, i.e. 2 more numbers. Please add.

"We changed the chart." How? Why do the original and its update figure look so different?

legend: "Chart 1. Correlations between birth weight and uric acid levels in the < 3rd PC group."

It does show a scatterplot and a linear regression line. Not "Correlations" (which should read correlation, anyway). Please change correspondingly.

" Those independent variables are included: UA-PI, Mean UtA-PI and sFlt-1/PlGF ratio." This needs to be properly explained in methods and not in a short reply only. Where are the corresponding regression results? Are you referring with these models by concluding remark "The application of an algorithm for low birth weight prognosis that makes use of 303 biophysical (ultrasound) and biochemical (uric acid level, angiogenesis markers) pa-304 rameters yields better results than using these parameters separately of each other."?

conclusions: "There is a significantly statistical" should read "significant"

Abstract; "(4)Conclusions: There is a strong link between" change to "significant" as in conclusions

Author Response

Point 1:  

Being statistician for many years I am quite aware of the compatibility between statistical tests and CIs. Nevertheless, as e.g. some known journals are requesting the use of confidence intervals only, a confidence interval enables a much better interpretation than a p-value does since it allows to quantify the effect. Table 1 for sure allows adding corresponding CIs for the differences, i.e. 2 more numbers. Please add.

Response 1: We have added 95%CI in the Table 1.

Point 2: 

How? Why do the original and its update figure look so different?

legend: "Chart 1. Correlations between birth weight and uric acid levels in the < 3rd PC group."

It does show a scatterplot and a linear regression line. Not "Correlations" (which should read correlation, anyway). Please change correspondingly.

Response 2: We took into account too few patients when creating this chart. It was a simple mistake. Thank You very much for this comment.

Point 3:  " Those independent variables are included: UA-PI, Mean UtA-PI and sFlt-1/PlGF ratio." This needs to be properly explained in methods and not in a short reply only. Where are the corresponding regression results? Are you referring with these models by concluding remark "The application of an algorithm for low birth weight prognosis that makes use of 303 biophysical (ultrasound) and biochemical (uric acid level, angiogenesis markers) pa-304 rameters yields better results than using these parameters separately of each other."?

Response 3:  We write in methods that  "Patients diagnosed with placental insufficiency were monitored according to generally accepted standards using biochemical, biophysical, and general condition parameters. Additionally, the angiogenesis markers sFlt-1 and PlGF, as well as uric acid levels, were determined in all patients."

We explain the corresponding regression results in Results - table 6.